# A clinical specific BERT developed using a huge Japanese clinical text corpus

Yoshimasa Kawazoe[1]☉*, Daisaku Shibata[1]☉, Emiko Shinohara[1]☉, Eiji Aramaki[2]☉, Kazuhiko Ohe[3]☉

**1** Artificial Intelligence in Healthcare, Graduate School of Medicine, The University of Tokyo, Tokyo, Japan, **2** Social Computing Lab, Graduate School of Information Science, Nara Institute of Science and Technology, Nara, Japan, **3** Department of Biomedical Informatics, Graduate School of Medicine, The University of Tokyo, Tokyo, Japan

☉ These authors contributed equally to this work.
* kawazoe@m.u-tokyo.ac.jp

**Data Availability Statement:** 1. The UTH-BERT model used for the experiment is available at our web site under the Creative Commons 4.0 International License (CC BY-NC-SA 4.0). URL: https://ai-health.m.u-tokyo.ac.jp/uth-bert 2. The

## Abstract

Generalized language models that are pre-trained with a large corpus have achieved great performance on natural language tasks. While many pre-trained transformers for English are published, few models are available for Japanese text, especially in clinical medicine. In this work, we demonstrate the development of a clinical specific BERT model with a huge amount of Japanese clinical text and evaluate it on the NTCIR-13 MedWeb that has fake Twitter messages regarding medical concerns with eight labels. Approximately 120 million clinical texts stored at the University of Tokyo Hospital were used as our dataset. The BERT-base was pre-trained using the entire dataset and a vocabulary including 25,000 tokens. The pre-training was almost saturated at about 4 epochs, and the accuracies of Masked-LM and Next Sentence Prediction were 0.773 and 0.975, respectively. The developed BERT did not show significantly higher performance on the MedWeb task than the other BERT models that were pre-trained with Japanese Wikipedia text. The advantage of pre-training on clinical text may become apparent in more complex tasks on actual clinical text, and such an evaluation set needs to be developed.

## 1 Introduction

In recent years, generalized language models that perform pre-training on a huge corpus have achieved great performance on a variety of natural language tasks. These language models are based on the transformer architecture, which is a novel neural network based solely on a self-attention mechanism [1]. Models such as Bidirectional Encoder Representations from Transformers (BERT) [2], Transformer-XL [3], XLNet [4], RoBERTa [5], XLM [6], GPT [7], and GPT-2 [8] have been developed and achieved state-of-the-art results. It is preferred that the domain of the corpus used for pre-training is the same as that of the target task. In the fields of life science and clinical medicine, domain-specific pre-trained models, such as Sci-BERT [9], Bio-BERT [10], and Clinical-BERT [11], have been published for English texts. A study that used the domain-specific pre-trained Clinical-BERT model yielded performance improvements on the tasks of common clinical natural language processing (NLP) compared to non-specific models.

NTCIR-13 dataset used for the experiment is available at the NTCIR web site upon request under the Creative Commons 4.0 International License (CC BY 4.0). URL: http://research.nii.ac.jp/ntcir/index-en.html 3. The KU-BERT model used for the experiment is available at the web site of Kurohashi-Chu-Murakami Lab, Kyoto University. URL: https://nlp.ist.i.kyoto-u.ac.jp/?ku_bert_japanese 4. The TU-BERT model used for the experiment is available at the web site of Tohoku NLP lab, Tohoku University. URL: https://github.com/cl-tohoku/bert-japanese.

**Funding:** This project was partly funded by the Japan Science and Technology Agency, Promoting Individual Research to Nurture the Seeds of Future Innovation and Organizing Unique, Innovative Network (JPMJPR1654). There were no other funders. The funders had no role in study design, data collection and analysis, decision to publish, or preparation of the manuscript.

**Competing interests:** Y.K, D.S, and E.S belong to the 'Artificial Intelligence in Healthcare, Graduate School of Medicine, The University of Tokyo' which is an endowment department, supported with an unrestricted grant from 'I&H Co., Ltd.' and 'EM SYSTEMS company', but these sponsors had no control over the interpretation, writing, or publication of this work. This does not alter our adherence to PLOS ONE policies on sharing data and materials.

While many BERT models for English have been published, few models are available for Japanese texts, especially in clinical medicine. One option available for Japanese clinical texts is the multilingual BERT (mBERT) published by Google; however, mBERT would have a disadvantage in word-based tasks because of its character-based vocabulary. For general Japanese texts, BERTs that have been pre-trained using Japanese Wikipedia have been published [12, 13]; however, their applicability to the NLP task for clinical medicine has not yet been studied. Because clinical narratives (physicians' or nurses' notes) have differences in linguistic characteristics from text on the web, pre-training on clinical text would be advantageous for the clinical NLP tasks. In this work, we developed and publicly released a BERT that was pre-trained with huge amount of Japanese clinical narratives. We also present the evaluation of the developed clinical-specific BERT through its comparison with three nonspecific BERTs for Japanese text based on a shared NLP task.

## 2 Methods

### 2.1 Datasets

Approximately 120 million lines of clinical text gathered over a period of eight years and stored in the electronic health record system of the University of Tokyo Hospital were used. Those texts were mainly recorded by physicians and nurses during daily clinical practice. Because Japanese text includes two-byte full-width characters (mainly Kanji, Hiragana, or Katakana) and one-byte half-width characters (mainly ASCII characters), the Normalization Form Compatibility Composition (NFKC) followed by full-width characterization were applied to all characters as a pre-processing task. Because the clinical text may contain personal information of patients, it was anonymized as much as possible by computer processing. Data collection followed a protocol approved by the Institutional Review Board (IRB) at the University of Tokyo Hospital (2019276NI). The IRB approved the possible inclusion of personal information in some of the texts used in this study.

### 2.2 Tokenization of Japanese text

To input a sentence into BERT, it is necessary to segment a sentence into tokens included in the vocabulary of BERT. In non-segmented languages such as Japanese or Chinese, a tokenizer must accurately identify every word in a sentence that requires a method of finding word boundaries without the aid of word delimiters. To obtain BERT tokens from Japanese text, morphological analysis followed by wordpiece tokenization was applied. Morphological analyzers such as MeCab [14] or Juman++ [15] is commonly used in Japanese text processing to segment a source text into word units that are pre-defined in its own dictionary. Subsequently, the wordpiece tokenization would be applied, which segments a word unit into several pieces of tokens included in the BERT vocabulary. During the wordpiece tokenization, a word like *playing* is segmented to two subwords, namely *play* and *##ing*. A subword that starts with ## represents a subword that is an appendage to another word. Fig 1 shows a schematic view of the morphological analysis and wordpiece tokenization of a Japanese text.

### 2.3 Making BERT vocabulary

A BERT model requires a fixed number of token vocabulary for wordpiece embeddings. To make the BERT vocabulary, candidate word pieces were obtained by applying morphological analysis followed by Byte Pair Encoding (BPE) [16] to the entire dataset. MeCab was used as a morphological analyzer along with the mecab-ipadic-NEologd dictionary [17] and the J-MeDic [18] as an external dictionary. The former had been built utilizing various resources

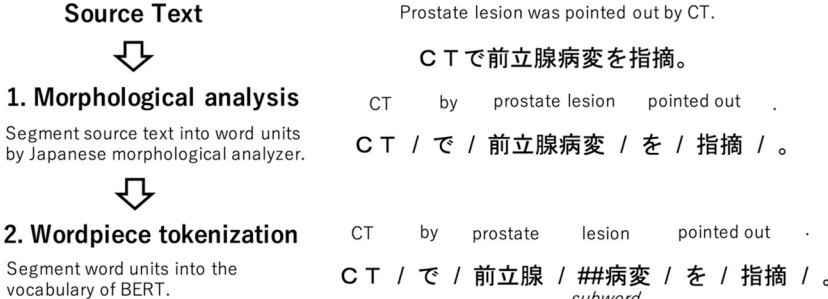

**Fig 1. The schematic view of morphological analysis and wordpiece tokenization.**

on the web, and it was used to identify personal names in clinical text as much as possible and aggregate them into a special token (@@N). The latter is a domain specific dictionary that had been built from Japanese clinical text, and it was used to segment words for diseases or findings into as large a unit as possible. BPE first decomposes a word unit into character symbols and, subsequently, creates a new symbol by merging two adjacent and highly frequent symbols. The merging process is stopped if the number of different symbols reaches the desired vocabulary size. In addition to this process, candidate words that represented specific people or facilities were excluded through manual screening, which allowed us to make the developed BERT publicly available. Eventually, 25,000 tokens including special tokens were adopted as the vocabulary.

## 2.4 Pre-training of BERT

BERT has shown state-of-the-art results for a wide range of tasks, such as single sentence classification, sentence pair classification, and question answering without substantial modifications to task specific architecture. The novelty of BERT is that it took the idea of learning word embeddings one step further, by learning each embedding vector considering the co-occurrence of words. To do this, BERT utilizes the self-attention mechanism, which learns sentence and word embeddings by capturing co-occurrence relationships between those embeddings. BERT is pre-trained by inputting fixed-length tokens obtained from two sentences and optimizing the Masked-LM and the Next Sentence Prediction simultaneously. As these two tasks do not require manually supervised labels, the pre-training is conducted as self-supervised learning.

## 2.5 Masked-LM

Fig 2A shows a schematic view of Masked-LM. This task masks, randomly replaces, or keeps each input token with a certain probability, and estimates the original tokens. Estimating not only the masked tokens but also the replaced or kept tokens help to keep a distributional contextual representation of every input token. Although the selection probability of the tokens to be dealt with is arbitrary, we used the 15% mentioned in the original paper [2].

## 2.6 Next Sentence Prediction

Fig 2B shows a schematic view of Next Sentence Prediction. In this task, the model receives pairs of sentences and predicts whether the second sentence of the pair is a consecutive sentence in the original dataset. To develop such a training dataset, for two consecutive sentences in the original dataset, the first sentence is connected to the original second sentence with a probability of 50% as a positive example. The remaining 50% of the time, the first sentence is

**A. Masked LM**

For each token, following action is
conducted by a 15 percent chance.

安静 / **加療** / 目的 / に / [UNK] / 入院 / 。

80%: Replace with *mask* token.
→ 安静 / **[MASK]** / 目的 / に / [UNK] / 入院 / 。

10%: Replace with randomly selected token.
→ 安静 / **外来** / 目的 / に / [UNK] / 入院 / 。

10%: Keep the token.
→ 安静 / **加療** / 目的 / に / [UNK] / 入院 / 。

Predict the original tokens for the masked,
replaced or kept tokens.

**B. Next sentence prediction**

Original two sentences appears in the dataset.

安静 / 加療 / 目的 / に / [UNK] / 入院 / 。入院 / 後 / 、
/ 安静 / ・ / 降圧 / にて / 経過 / 観察 / 。

50%: The subsequent sentence is kept.
**[CLS]** / 安静 / [MASK] / 目的 / に / [UNK] / 入院 / 。 /
→ **[SEP]** / 入院 / 後 / 、 / 安静 / ・ / [MASK] / にて / 経
過 / 観察 / 。 / **[SEP]**

50%: The subsequent sentence is replaced with a randomly
sampled sentence.
**[CLS]** / 安静 / [MASK] / 目的 / に / [UNK] / 入院 / 。 /
→ **[SEP]** / *C T* / で / *前立腺* / *##病変* / *[MASK]* / *指摘*
/ 。 / **[SEP]**

Predict if the second sentence in the pair is the
subsequent in the dataset.

**Fig 2. The schematic view of Masked-LM and Next Sentence Prediction task. A.** Masked LM predicts the original
tokens for the masked, replaced or kept tokens. **B.** Next Sentence Prediction predicts if the second sentence in the pair
is the subsequent sentence in the original documents. The role of special symbols are as follows: [CLS] is added in front
of every input text, and the output vector is used for Next Sentence Prediction task; [MASK] is masked token in
Masked-LM task; [SEP] is a break between sentences; [UNK] is unknown token that does not appear in the
vocabulary.

connected to a randomly sampled sentence as negative example. We treated all sentences
appearing in a document recorded in one day for a patient as consecutive sentences.

## 2.7 Evaluation task

The performance of the developed BERT was evaluated through a fine-tuned approach using
the NTCIR-13 Medical Natural Language Processing for Web Document (MedWeb) task [19].
MedWeb is publicly available and provides manually created fake Twitter messages regarding
medical concerns in a cross-language and multi-label corpus, covering three languages (Japa-
nese, English, and Chinese), and annotated with eight labels. A Positive or Negative status is
given to eight labels of Influenza, Diarrhea, Hay fever, Cough, Headache, Fever, Runny nose,
and Cold; the Positive status may be given to multiple labels in a message. We performed a
multi-label task to classify these eight classes simultaneously. Table 1 shows examples of each
set of pseudo-tweets.

## 2.8 Experimental settings

For the pre-training experiments, we leveraged the Tensorflow implementation of BERT-base
(12 layers, 12 attention heads, 768 embedding dimension, 110 million parameters) published
by Google [2]. Approximately 99% of the 120 million sentences was used for training, and the
remaining of 1% was used for the evaluation of the accuracies of Masked LM and Next Sen-
tence Prediction. For the evaluation experiments, the pre-trained BERT was fine-tuned. The
network was configured such that the output vector C corresponding to the first input token
([CLS]) was linearly transformed to eight labels by a fully connected layer, and the Positive or
Negative status of each of the eight labels were outputted through a sigmoid function. Binary
cross entropy was used for the loss function, and the learning rate was optimized by Adam ini-
tialized with 1e-5. All network parameters including BERT were updated during this fine-tun-
ing process. Fig 3 shows a schematic view of this network. Five models were trained by 5-fold
cross-validation using the MedWeb training data consisting of 1,920 texts, and the mean
results of the models on the MedWeb test data consisting of 640 texts were assessed. The per-
formance was assessed based on the exact-match accuracy and label-wise F-measure (macro

**Table 1. Three examples of pseudo-tweets with the eight classes of symptoms.**

| | Lang | Pseudo-tweets | Flu | Diarrhea | Hay fever | Cough | Headache | Fever | Runny nose | Cold |
|---|---|---|---|---|---|---|---|---|---|---|
| 1 | ja | 風邪で鼻づまりがやばい。 | N | N | N | N | N | N | P | P |
| | en | I have a cold, which makes my nose stuffy like crazy. | | | | | | | | |
| | zh | 感冒引起的鼻塞很烦人。 | | | | | | | | |
| 2 | ja | 花粉症のせいでずっと微熱でぼーっとしてる。眠い。 | N | N | P | N | N | P | P | N |
| | en | I'm so feverish and out of it because of my allergies. I'm so sleepy. | | | | | | | | |
| | zh | 由于花粉症一直发低烧，晕晕沉沉的。很困。 | | | | | | | | |
| 3 | ja | 鼻風邪かなと思ってたけど、頭痛もしてきたから今日は休むことにしよう。 | N | N | N | N | P | N | P | P |
| | en | It was just a cold and a runny nose, but now my head is starting to hurt, so I'm gonna take a day off today. | | | | | | | | |
| | zh | 想着或许是鼻伤风，可头也开始疼了，所以今天就休息吧。 | | | | | | | | |

The English (en) and Chinese (zh) sentences were translated from Japanese (ja).

F1). To inspect an advantage of the domain specific model, we also evaluated the two kinds of domain nonspecific BERT that are pre-trained in Japanese Wikipedia and mBERT. Table 2 shows the specifications of each BERT model.

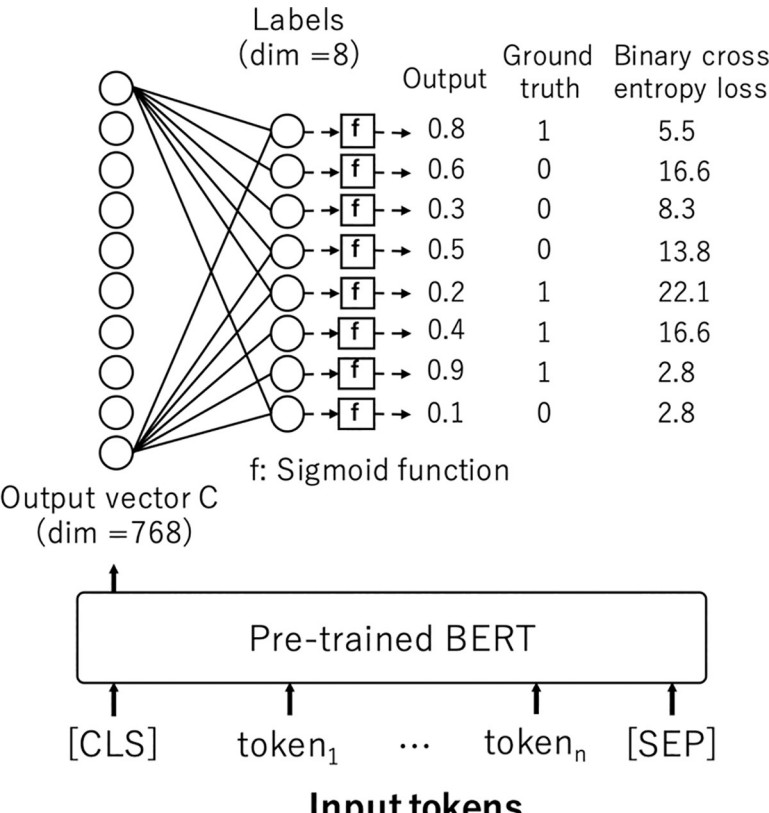

**Fig 3. The schematic view of the network for evaluation.**

**Table 2. The specifications of each BERT.**

| | | UTH-BERT | KU-BERT | TU-BERT | mBERT |
|---|---|---|---|---|---|
| Publisher | | The University of Tokyo Hospital | The University of Kyoto | The University of Tohoku | Google |
| Language | | Japanese | Japanese | Japanese | Multilingual |
| Pre-training corpus | | Clinical text (120 million) | JP Wikipedia (18 million) | JP Wikipedia (18 million) | 104 languages of Wikipedias |
| Tokenizer | Morphological analyzer | MeCab | Juman++ | MeCab | - |
| | External Dictionary | Mecab-ipadic-neologd, J-MeDic | - | Mecab-ipadic | - |
| Number of vocabularies | | 25,000 | 32,000 | 32,000 | 119,448 |
| Total number of [UNK] tokens present in the MedWeb dataset. | | 253 (0.68%) | 394 (1.11%) | 369 (0.94%) | 1 (0.00%) |

## 3 Results

### 3.1 Pre-training performance

Table 3 shows the results of the pre-training. The pre-training was almost saturated at approximately 10 million steps (4 epochs), and the accuracies of Masked LM and Next Sentence Prediction were 0.773 and 0.975, respectively. With a mini-batch size of 50, 2.5 million steps are equivalent to approximately 1 epoch. It took approximately 45 days to learn 4 epochs using a single GPU. In the subsequent experiment, UTH-BERT with 10 million steps of training was used.

### 3.2 Finetuning performance

Table 4 shows the exact-match accuracies with 95% confidence intervals of four pre-trained BERTs. There were no significant differences among UTH-BERT (0.855), KU-BERT (0.845), and TU-BERT (0.862); however, mBERT significantly showed the lowest accuracy compared to the other BERTs.

Table 5 shows the label-wise Recall, Precision, and F-measure of each model. There were no significant differences in the mean F-measures among UTH-BERT (0.888), KU-BERT (0.882), TU-BERT (0.888), and mBERT (0.855). The mean F-measure of mBERT tended to be lower than other BERT models, but the difference was not significant. In terms of the performance for each symptom, the mean F values for Flu (0.714) and Fever (0.838) were lower than for the other symptoms.

### 3.3 Error analysis

To obtain a better understanding of the UTH-BERT classifier's mistakes, we qualitatively analyzed its false positive (FP) and false negative (FN) cases in the 640 MedWeb test dataset. The error analysis was conducted for the labels for which UTH-BERT was wrong all five times in the five-fold cross validation. The labels of MedWeb were annotated in terms of three aspects such as *Factuality* (whether the tweeter has certain symptom or not), *Location* (whether the symptoms are those of the tweeter or someone in the vicinity or not), and *Tense* (whether the symptoms exist within 24 hours or not) [19]. Since the MedWeb dataset did not contain

**Table 3. Accuracies of Masked-LM and Next Sentence Prediction in pre-training for the evaluation dataset.**

| UTH-BERT | Number of training steps (epochs) | | | |
|---|---|---|---|---|
| | $2.5 \times 10^6$ (1) | $5.0 \times 10^6$ (2) | $7.5 \times 10^6$ (3) | $10 \times 10^6$ (4) |
| Masked LM (accuracy) | 0.743 | 0.758 | 0.768 | 0.773 |
| Next Sentence Prediction (accuracy) | 0.966 | 0.970 | 0.973 | 0.975 |

**Table 4. The exact-match accuracy of each model with five-fold cross validation.**

| Model name | Exact match accuracy (95% CI) |
|---|---|
| UTH-BERT | 0.855 (0.848–0.862) |
| KU-BERT | 0.845 (0.833–0.857) |
| TU-BERT | 0.862 (0.857–0.866) |
| mBERT | 0.806 (0.794–0.817) |

information about these perspectives, we manually categorized the error cases based on these aspects. As a result, we obtained eight error types for the FP cases and five error types for the FN cases (Table 6).

**1. FP due to false detection of co-occurring symptoms.** This type of error was categorized as *Factuality*. The example sentence No.1 expresses that the flu is negative, but UTH--BERT incorrectly predicted that the fever and flu are positive. The reason for this error would be that the training data contained sentences in which flu and fever are positive simultaneously. In addition, UTH-BERT could not detect the negative expression of flu, so both flu and fever were incorrectly positive.

**2. FP for symptoms mentioned in general topics.** This was categorized as *Factuality*. The example sentence No. 2 states that a runny nose is a common symptom of a cold. Despite the general topic, UTH-BERT incorrectly predicted that the tweeter has a cold and a runny nose. The reason for this error would be that UTH-BERT failed to distinguish between symptoms that are stated as a general topic and those occurring in a person.

**3. FP for suspected influenza.** This was categorized as *Factuality*. According to the Med-Web annotation criteria, suspected symptoms were treated as positive, but only influenza was treated as negative. (This is because the MedWeb dataset was developed primarily for the surveillance of influenza.) This suggests that difference of annotation criteria between flu and other symptoms, and a lack of sentence expression about suspected flu in the training dataset, led to the errors.

**4. FP for fully recovered symptoms.** This was categorized as *Factuality*. According to the annotation criteria, symptoms are labeled as positive if they are in the recovery process and negative if they are completely cured. In example sentence No. 4, even though the tweeter stated that the cough was cured, UTH-BERT could not recognize *cured* and incorrectly predicted that cough was positive.

**Table 5. The label-wise performances of each model with five-fold cross validation.**

| | Flu | Diarrhea | Hay fever | Cough | Headache | Fever | Runny nose | Cold | *Mean F1 (95% CI)* |
|---|---|---|---|---|---|---|---|---|---|
| | *Rec. / Prec.* | *Rec. / Prec.* | *Rec. / Prec.* | *Rec. / Prec.* | *Rec. / Prec.* | *Rec. / Prec.* | *Rec. / Prec.* | *Rec. / Prec.* | |
| | *F1* | *F1* | *F1* | *F1* | *F1* | *F1* | *F1* | *F1* | |
| UTH | 0.676/0.858 | 0.914/0.919 | 0.904/0.835 | 0.928/0.963 | 0.947/0.974 | 0.797/0.905 | 0.920/0.927 | 0.885/0.913 | 0.888 (0.846–0.931) |
| | **0.755** | 0.916 | 0.865 | **0.945** | **0.960** | 0.845 | 0.923 | 0.898 | |
| KU | 0.594/0.842 | 0.877/0.956 | 0.896/0.896 | 0.892/0.963 | 0.947/0.958 | 0.760/0.927 | 0.921/0.927 | 0.890/0.936 | 0.882 (0.828–0.935) |
| | 0.694 | 0.915 | **0.895** | 0.926 | 0.952 | 0.835 | 0.924 | **0.912** | |
| TU | 0.735/0.692 | 0.927/0.947 | 0.898/0.874 | 0.916/0.957 | 0.936/0.982 | 0.825/0.903 | 0.912/0.938 | 0.884/0.904 | 0.888 (0.837–0.939) |
| | 0.710 | **0.937** | 0.885 | 0.936 | 0.958 | **0.861** | **0.925** | 0.893 | |
| mBERT | 0.598/0.850 | 0.867/0.906 | 0.870/0.822 | 0.918/0.892 | 0.928/0.927 | 0.745/0.890 | 0.869/0.893 | 0.902/0.887 | 0.855 (0.807–0.902) |
| | 0.696 | 0.885 | 0.841 | 0.905 | 0.926 | 0.810 | 0.879 | 0.894 | |
| *Mean F1* | 0.714 | 0.913 | 0.872 | 0.928 | 0.949 | 0.838 | 0.913 | 0.899 | |

The performances shown are Recall, Precision and F-measure.

**Table 6. Interpretations obtained from the results of the error analysis.**

| No. | Error | Cause of the error | Num. of errors | Example sentence | Incorrect prediction |
|---|---|---|---|---|---|
| 1 | False positive (FP) | Co-occurring symptoms | 10 | (ja) インフルかと思って病院行ったけど、検査したら違ったよ。<br>(en) I thought I had the flu so I went to the doctor, but I got tested and I was wrong. | Fever pos. |
| 2 | | Symptoms mentioned in general topics | 8 | (ja) 風邪といえば鼻づまりですよね。<br>(en) To me, a cold means a stuffy nose. | Cold pos. Runny nose pos. |
| 3 | | Suspected influenza | 5 | (ja) インフルエンザかもしれないから部活休もうかな。<br>(en) I might have the flu so I'm thinking I'll skip the club meeting. | Flu pos. |
| 4 | | Fully recovered symptoms | 5 | (ja) やっと咳と痰が治まった。<br>(en) My cough and phlegm are finally cured. | Cough pos. |
| 5 | | Metaphorical expressions | 3 | (ja) 熱をあげているのは嫁と娘だ。<br>(en) What makes me excited are my wife and daughter. | Fever pos. |
| 6 | | Denied symptoms | 2 | (ja) 鼻水が止まらないので熱でもあるのかと思ったけど、全然そんなことなかったわ。<br>(en) My nose won't stop running, which got me wondering if I have a fever, but as it turns out I definitely do not. | Fever pos. |
| 7 | | Symptoms for asking unspecified people | 2 | (ja) 誰か熱ある人いない?<br>(en) Anyone have a fever? | Fever pos. |
| 8 | | Past symptoms | 1 | (ja) ネパールにいったら食べ物があわなくてお腹壊して下痢になった・・・<br>(en) When I went to Nepal, the food didn't agree with me, and I got an upset stomach and diarrhea. . . | Diarrhea pos. |
| 9 | False negative (FN) | Symptoms that are directly expressed | 8 | (ja) 痰が止まったとおもったらこんどは頭痛。<br>(en) Just when I thought the phlegm was over, now I have a headache | Headache neg. |
| 10 | | Symptoms that are indirectly expressed | 5 | (ja) 中国にいた時は花粉症ならなかったのに再発したー!<br>(en) Even though I didn't have allergies when I was in China, they're back! | Runny nose neg. |
| 11 | | Symptoms that can be inferred to be positive by being a tweet from a person | 4 | (ja) 今日花粉少ないとか言ってるやつ花粉症じゃないから。<br>(en) The people who are saying there's not a lot of pollen today don't have allergies. | Runny nose neg. |
| 12 | | Symptoms that are in the recovery process | 1 | (ja) インフルが回復してきてだいぶ元気になった!けどあと2日は外出禁止なんだよな。<br>(en) I've recovered from the flu and feel great! But I'm still not allowed to go out for two days. | Flu neg. |
| 13 | | Symptoms occurring in the tweeter's neighborhood | 1 | (ja) うちのクラス、集団で下痢事件<br>(en) There's a diarrhea outbreak in my class | Diarrhea neg. |

**5. FP for metaphorical expressions.** This was categorized as *Factuality*. This error is due to an inability to recognize metaphorical expressions. In example sentence No. 5, the Japanese phrase熱を上げる is a metaphorical expression for *excited*, but because it uses the same kanji as *fever*, UTH-BERT incorrectly predicted that fever is positive.

**6. FP for denied symptoms.** This was categorized as *Factuality*. This error is caused by UTH-BERT missing a negative expression.

**7. FP for symptoms for asking unspecified people.** This was categorized as *Location*. Although example sentence No. 7 asks about the presence of fever for an unspecified person, UTH-BERT incorrectly predicted that the tweeter has a fever.

**8. FP for past symptoms.**    This was categorized as *Tense*. According to the annotation criteria, past symptoms are treated as negative. This error occurred because UTH-BERT was not able to recognize the tense.

**9. FN for symptoms that is directly expressed.**    This was categorized as *Factuality*. Although the sentences directly express that the tweeter has a symptom, this type of errors occurred because UTH-BERT could not detect it.

**10. FN for symptoms that is indirectly expressed.**    This was categorized as *Factuality*. This is the type of error that overlooks a symptom that would be inferred to be positive if another symptom was present at the same time. Example sentence No.10 directly expresses that hay fever is positive, but if you have some knowledge, you can guess that runny nose is also positive.

**11. FN for symptoms that can be inferred to be positive because the tweet is from a person.**    This was categorized as *Factuality*. This is also the type of error that overlooks a symptom that is expressed indirectly, but it requires more advanced reasoning. Example sentence No.11 states a general topic, but given that it is a tweet, you can infer that the person has hay fever.

**12. FN for symptoms that is in the recovery process.**    This was categorized as *Factuality*. According to the annotation criteria, symptoms are labeled as positive if they are in the recovery process; however, UTH-BERT could not detect it.

**13. FN for symptoms occurring in the tweeter's neighborhood.**    This was categorized as *Location*. According to the annotation criteria, if a population in the same space has a symptom, the symptom is annotated as positive regardless of whether the tweeter has the symptom or not. In this case, UTH-BERT predicted it as negative, because there were probably not enough such cases in the training data.

## 4 Discussions

We presented a BERT model pre-trained with a huge amount of Japanese clinical text and evaluated it on the NTCIR-13 MedWeb task. To the best of our knowledge, this work is the first to inspect a BERT model that is pre-trained using Japanese clinical text and publish the results. Among the BERT models, UTH-BERT, KU-BERT, and TU-BERT, which are specialized for Japanese text, significantly outperformed mBERT in exact-match accuracy and tended to outperform mBERT in label-wise accuracy. The mBERT uses character-based vocabulary that alleviates vocabulary problems of handling multiple languages instead of giving up the semantic information that words have. This result would indicate a disadvantage for character-based vocabulary compared to word-based vocabulary. Nevertheless, the performance of mBERT was close to the other Japanese BERT models. Regarding the advantages of pre-training with clinical text, UTH-BERT showed no significant advantages over KU-BERT and TU-BERT. One of the reasons is that sentence classification is a relatively easy task for BERTs that are pre-trained on a large text corpus; therefore, the advantage of pre-training with the domain text may not have been noticeable. Further, because the NTCIR-13 MedWeb used for the evaluation is an intermediate corpus between web and medicine, the differences among the BERTs may not have been clear. The advantage of training on domain-specific texts may become apparent in more complex tasks such as named entity recognition, relation extraction, question answering, or causality inference on clinical text, and a Japanese corpus for such an evaluation is yet to be developed.

We conducted an error analysis that resulted in 13 different types of error interpretations. Among these interpretations, errors related to the factuality of symptoms were the most common, and errors related to the location and the tense were less common. This bias could be

due to the small amount of data labeled for location and tense in the MedWeb dataset, rather than a feature of the UTH-BERT. The most common error type in the FP cases was due to false detection of co-occurring symptoms, and 10 of these errors were found in this analysis. Since the task in the MedWeb dataset is a multi-label classification of eight symptoms, this error would be influenced by co-occurrence relationships of multiple labels appearing in the training dataset. On the other hand, some of the FN error types were found to overlook symptom that were expressed indirectly. A possible way to reduce such oversights would be to prepare many similar cases in the training dataset, but it seemed to be difficult as long as only text was used as a source of information. It was difficult to conduct further analysis since this error analysis was based on manual categorization. For further investigation, it would be possible to apply Shapley additive explanations (SHAP) [20] or local interpretable model-agnostic explanations (LIME) [21] to visualize the effect of the predictions when the input data is perturbed by deleting or replacing input tokens.

Differences in the distribution of words appearing in the clinical text used for pre-training of UTH-BERT and the pseudo-tweet messages used for evaluation may have affected the errors. Clinical texts differ in that they contain objective information about the patient, while pseudo-tweets message contain subjective information about the tweeter. Another difference is that the former is written in a literary style, while the latter is written in a spoken form. Because of these differences, there may be cases where the representation of the UTH-BERT pre-acquired tokens were not maximally utilized in the pseudo-tweet messages, leading to errors. A limitation of our error analysis is that it was not possible to compare the error trends between the BERT models because we did not perform the error analysis for KU-BERT and TU-BERT. Moreover, given that our developed BERT was evaluated exclusively on the NTCIR-13 MedWeb task, there is currently a limitation in the generalizability of the performance.

## 5 Conclusions

We developed a BERT model that made use of a huge amount of Japanese clinical text and evaluated it on the NTCIR-13 MedWeb dataset to investigate the advantage of a domain-specific model. The result shows that there are no significant differences among the performances of BERT models that are pre-trained with Japanese text. Our aim is to develop publicly available tools that will be useful for NLP in the clinical domain; however, knowing the nature of the developed model require evaluations based on more complex tasks such as named entity recognition, relation extraction, question answering, and causality inference on actual clinical text.

## Author Contributions

**Conceptualization:** Yoshimasa Kawazoe, Emiko Shinohara.

**Investigation:** Daisaku Shibata.

**Methodology:** Yoshimasa Kawazoe, Daisaku Shibata.

**Resources:** Emiko Shinohara, Kazuhiko Ohe.

**Supervision:** Eiji Aramaki, Kazuhiko Ohe.

**Writing – original draft:** Yoshimasa Kawazoe.

**Writing – review & editing:** Yoshimasa Kawazoe, Eiji Aramaki.

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
