## [Decision Letter · Decision Letter 0]

3 Dec 2020

PONE-D-20-20418

A clinical specific BERT developed with huge size of Japanese clinical narrative

PLOS ONE

Dear Dr. Kawazoe,

Thank you for submitting your manuscript to PLOS ONE. After careful consideration, we feel that it has merit but does not fully meet PLOS ONE’s publication criteria as it currently stands. Therefore, we invite you to submit a revised version of the manuscript that addresses the points raised during the review process.

We look forward to receiving your revised manuscript.

Kind regards,

Diego Raphael Amancio

Academic Editor

PLOS ONE

Journal Requirements:

2. In the ethics statement in the manuscript and in the online submission form, please provide additional information about the patient records used in your retrospective study. Specifically, please ensure that you have discussed whether all data were fully anonymized before you accessed them and/or whether the IRB or ethics committee waived the requirement for informed consent. If patients provided informed written consent to have data from their medical records used in research, please include this information.

"Y.K and E.S belong to the 'Artificial Intelligence in Healthcare, Graduate School of Medicine, The University of Tokyo' which is an endowment department, supported with an unrestricted grant from ‘I&H Co., Ltd.’ and ‘EM SYSTEMS company’, but these sponsors had no control over the interpretation, writing, or publication of this work."

Reviewers' comments:

Reviewer's Responses to Questions

**Comments to the Author**

1. Is the manuscript technically sound, and do the data support the conclusions?

Reviewer #1: Yes

Reviewer #2: Partly

2. Has the statistical analysis been performed appropriately and rigorously? 

Reviewer #1: Yes

Reviewer #2: Yes

3. Have the authors made all data underlying the findings in their manuscript fully available?

Reviewer #1: Yes

Reviewer #2: Yes

4. Is the manuscript presented in an intelligible fashion and written in standard English?

Reviewer #1: Yes

Reviewer #2: No

5. Review Comments to the Author

Reviewer #1: Report on the manuscript "A clinical specific BERT developed with huge size of

Japanese clinical narrative" by Kawazoe and coauthors submitted for

publication in PLOS ONE.

In this manuscript, the authors present a clinical specific BERT model trained

on a massive data set comprising over 120 million lines of clinical text

obtained from the University of Tokyo Hospital. As the authors rightly point

out, there are very few models pre-trained with Japanese texts in general, and

particularly in the clinical domain. The authors compare its BERT model

(UTH-BERT, pre-trained with clinical text) with three other BERT models:

KU-BERT and TU-BERT (both pre-trained with the Japanese Wikipedia), and the

Google multilingual BERT (Google-ML). They observe that BERT models

pre-trained with Japanese texts outperform Google-ML, but no substantial

improvement is found between UTH-BERT and KU-BERT and TU-BERT (that is, there

is no significant advantage in using domain-specific texts). However, the

authors argue that this difference may emerge with more complex tasks.

This work reads very well, and I believe it is an essential contribution to

literature as it reduces the shortage of studies with the Japanese language

and may trigger other investigations. I have no suggestions on how to improve

this work, and I recommend publication in the present form.

Reviewer #2: This paper describes a clinical-specific BERT model for Japanese. The

model is pre-trained by using a huge Japanese clinical text. The

experiments on a text classification task in a clinical domain

demonstrate the proposed BERT model was slightly better than general

BERT models.

The trained BERT model is valuable for Japanese clinical

researches. However, the differences between the proposed BERT and

other general Japanese BERT models in the evaluated task are very

small. Although I understand the situation where only a few Japanese

evaluation sets in a clinical domain are available, this experimental

result is weak for insisting the effectiveness of the

clinical-specific Japanese BERT model.

Although the authors say that "error analysis is required" in L277,

improved examples and errors should be presented, and discussion

should be made in the paper. These will help readers understand the

strength and weakness of the proposed BERT model.

The comparison between BERT and other classical machine learning

methods such as SVM and LR does not make sense in this paper because

the superiority of BERT models has been shown in many

papers. Furthermore, SVM with KU or TU vocabularies does not make

sense because these (subword) vocabularies are not for SVM nor LR.

There are many typos and some misunderstandings for BERT. Please check

the followings carefully.

- L40: that pre-trained -> that are pre-trained

- L55: on huge corpus -> on a huge corpus

- L56: natural language task -> natural language tasks

- L60: corpus for pre-training preferred to use the same domain as the target task

-> the domain of a corpus for pre-training prefers to the same as the one of a target task

- L63: a study that domain specifically pre-training -> the domain-specific pre-trained model (?)

- L66: pre-trained transforms -> BERT models

- L67: Japanese text -> a Japanese text

- L67: Because multilingual BERT was pre-trained using a general

corpus, the sentence "One of the options is to use multilingual BERT

.." is not appropriate

- L70: WIKIPEDIA -> Wikipedia

- L73: make -> makes

- L90: I can't understand the sentence ".. before attempt to parse it .."

- L93: the MeCab -> MeCab, the Juman++ -> Juman

- L93: The reference [15] is wrong, and should be the following:

Morphological Analysis for Unsegmented Languages using Recurrent Neural Network Language Model.

Hajime Morita, Daisuke Kawahara, Sadao Kurohashi.

EMNLP 2015

- L94: wordpiece tokenization -> the wordpiece tokenization

- L95: which segment -> which segments

- L97: All the tokens are called "subword". The explanation should be as follows:

"A subword that starts with "##" represents a subword that is not the begging of a word."

- L107: external dictionary -> an external dictionary

- L109: the clinical text -> a clinical text

- L110: domain specific dictionary -> a domain specific dictionary

- L110: Japanese clinical text -> a Japanese clinical text

- L111: in as -> into as (?)

- L111: decompose -> decomposes

- L112: create -> creates

- L118: The paragraph "Pre-training BERT" contains several inaccurate expressions.

- L122: What does "the sequence" mean? It means a word sequence? The

word2vec embeddings, for example, are trained from a word sequence,

and so this explanation is inaccurate for mentioning the difference

between BERT and existing embeddings.

- L123: I think the clause "which learns sentence expressions

.. between words" is misunderstanding. The self-attention in BERT

can learn token embeddings as well as sentence embeddings.

- L131: the original embeddings of those tokens -> the original tokens

- L132: I can't understand the sentence "more appropriate

representation of sentences is obtained".

- L138: the consecutive sentence -> a consecutive sentence

- L140: The subject of the verb "connect" is missing

- L142: I think the verb "pinch" is not appropriate in this context

- L143: I think there is a misunderstanding in the NSP task. For a

sentence in a document, a random sentence as a negative example is

chosen from other documents. Therefore, it is a usual way that all

the sentences in a document in one day are regarded as one

document. The explanation from L142 to L147 does not make sense.

- L161: Please explain "pseudo-Twitter messages".

- Table2: Juman -> Juman++ (for KU-BERT)

- L203: single GPU -> a single GPU

- L211: "due to" is incorrect. I don't know the intention of this

phrase.

- L213: Google-ML BERT -> Google mBERT

- L225: highest -> the highest

- L237: by -> with

- L251: indicate -> indicates

- L256: which specialized to -> which are specialized to

- L258: use -> uses

- L258: alleviate -> alleviates

- L265: have pre-trained -> are pre-trained

- L267: intermediate corpus -> an intermediate domain

- L282: contribute in -> contribute to

6. PLOS authors have the option to publish the peer review history of their article (what does this mean?). If published, this will include your full peer review and any attached files.

Reviewer #1: No

Reviewer #2: **Yes: **Tomohide Shibata

---

## [Author Response · Author response to Decision Letter 0]

6 Jul 2021

Thank you for giving us the opportunity to submit a revised draft of the manuscript. We appreciate the time and effort that you dedicated to provide us with your valuable feedback on the manuscript. We are grateful to the reviewers for their insightful comments on our paper. We incorporated changes to reflect most of the suggestions provided by the reviewers.

At first, due to incorrect spelling, we would like to change the title as follows: A clinical specific BERT developed using a huge Japanese clinical text corpus.

Here is a point-by-point response to the reviewers’ comments and concerns.

# Reviewer1

1. This work reads very well, and I believe it is an essential contribution to literature as it reduces the shortage of studies with the Japanese language and may trigger other investigations. I have no suggestions on how to improve this work, and I recommend publication in the present form.

Response to 1

Thank you for your comment. We trust that the pre-trained model will help research on clinical texts.

# Reviewer2

1. The trained BERT model is valuable for Japanese clinical research. However, the differences between the proposed BERT and other general Japanese BERT models in the evaluated task are very small. Although I understand the situation where only a few Japanese evaluation sets in a clinical domain are available, this experimental result is weak for insisting the effectiveness of the clinical-specific Japanese BERT model.

Response to 1

Thank you for pointing this out. As a result of the re-experiment, we confirmed that there are no significant differences between the BERTs. In addition, because there are, to our knowledge, few Japanese datasets in the clinical domain to use in our experiments, evaluating the BERT models with an appropriate one was challenging. We modified our results as follows:

On Page 10,

Table 4 shows the exact-match accuracies with 95% confidence intervals of four pre-trained BERTs. There were no significant differences among UTH-BERT (0.855), KU-BERT (0.845), and TU-BERT (0.862); however, mBERT significantly showed the lowest accuracy compared to the other BERTs. 

2. Although the authors say that "error analysis is required" in L277, improved examples and errors should be presented, and discussion should be made in the paper. These will help readers understand the strength and weakness of the proposed BERT model.

Response to 2

Thank you for pointing this out. We agree that an error analysis is important to know the nature of the model. We conducted a qualitative error analysis and described it as follows:

On Page 12,

3.3 Error analysis

To obtain a better understanding of the UTH-BERT classifier’s mistakes, we qualitatively analyzed its false positive (FP) and false negative (FN) cases in the 640 MedWeb test dataset. The error analysis was conducted for the labels for which UTH-BERT was wrong all five times in the five-fold cross validation. The labels of MedWeb were annotated in terms of three aspects such as Factuality (whether the tweeter has certain symptom or not), Location (whether the symptoms are those of the tweeter or someone in the vicinity or not), and Tense (whether the symptoms exist within 24 hours or not) [19]. Since the MedWeb dataset did not contain information about these perspectives, we manually categorized the error cases based on these aspects. As a result, we obtained eight error types for the FP cases and five error types for the FN cases (Table 6).

1. FP due to false detection of co-occurring symptoms

This type of error was categorized as Factuality. The example sentence No.1 expresses that the flu is negative, but UTH-BERT incorrectly predicted that the fever and flu are positive. The reason for this error would be that the training data contained sentences in which flu and fever are positive simultaneously. In addition, UTH-BERT could not detect the negative expression of flu, so both flu and fever were incorrectly positive.

2. FP for symptoms mentioned in general topics

This was categorized as Factuality. The example sentence No. 2 states that a runny nose is a common symptom of a cold. Despite the general topic, UTH-BERT incorrectly predicted that the tweeter has a cold and a runny nose. The reason for this error would be that UTH-BERT failed to distinguish between symptoms that are stated as a general topic and those occurring in a person.

3. FP for suspected influenza

This was categorized as Factuality. According to the MedWeb annotation criteria, suspected symptoms were treated as positive, but only influenza was treated as negative. (This is because the MedWeb dataset was developed primarily for the surveillance of influenza.) This suggests that difference of annotation criteria between flu and other symptoms, and a lack of sentence expression about suspected flu in the training dataset, led to the errors.

4. FP for fully recovered symptoms

This was categorized as Factuality. According to the annotation criteria, symptoms are labeled as positive if they are in the recovery process and negative if they are completely cured. In example sentence No. 4, even though the tweeter stated that the cough was cured, UTH-BERT could not recognize cured and incorrectly predicted that cough was positive.

5. FP for metaphorical expressions

This was categorized as Factuality. This error is due to an inability to recognize metaphorical expressions. In example sentence No. 5, the Japanese phrase熱を上げる is a metaphorical expression for excited, but because it uses the same kanji as fever, UTH-BERT incorrectly predicted that fever is positive.

6. FP for denied symptoms

This was categorized as Factuality. This error is caused by UTH-BERT missing a negative expression. 

7. FP for symptoms for asking unspecified people

This was categorized as Location. Although example sentence No. 7 asks about the presence of fever for an unspecified person, UTH-BERT incorrectly predicted that the tweeter has a fever.

8. FP for past symptoms

This was categorized as Tense. According to the annotation criteria, past symptoms are treated as negative. This error occurred because UTH-BERT was not able to recognize the tense.

9. FN for symptoms that is directly expressed

This was categorized as Factuality. Although the sentences directly express that the tweeter has a symptom, this type of errors occurred because UTH-BERT could not detect it.

10. FN for symptoms that is indirectly expressed

This was categorized as Factuality. This is the type of error that overlooks a symptom that would be inferred to be positive if another symptom was present at the same time. Example sentence No.10 directly expresses that hay fever is positive, but if you have some knowledge, you can guess that runny nose is also positive.

11. FN for symptoms that can be inferred to be positive because the tweet is from a person.

This was categorized as Factuality. This is also the type of error that overlooks a symptom that is expressed indirectly, but it requires more advanced reasoning. Example sentence No.11 states a general topic, but given that it is a tweet, you can infer that the person has hay fever.

12. FN for symptoms that is in the recovery process

This was categorized as Factuality. According to the annotation criteria, symptoms are labeled as positive if they are in the recovery process; however, UTH-BERT could not detect it.

13. FN for symptoms occurring in the tweeter's neighborhood

This was categorized as Location. According to the annotation criteria, if a population in the same space has a symptom, the symptom is annotated as positive regardless of whether the tweeter has the symptom or not. In this case, UTH-BERT predicted it as negative, because there were probably not enough such cases in the training data.

On Page 18,

We conducted an error analysis that resulted in 13 different types of error interpretations. Among these interpretations, errors related to the factuality of symptoms were the most common, and errors related to the location and the tense were less common. This bias could be due to the small amount of data labeled for location and tense in the MedWeb dataset, rather than a feature of the UTH-BERT. The most common error type in the FP cases was due to false detection of co-occurring symptoms, and 10 of these errors were found in this analysis. Since the task in the MedWeb dataset is a multi-label classification of eight symptoms, this error would be influenced by co-occurrence relationships of multiple labels appearing in the training dataset. On the other hand, some of the FN error types were found to overlook symptom that were expressed indirectly. A possible way to reduce such oversights would be to prepare many similar cases in the training dataset, but it seemed to be difficult as long as only text was used as a source of information. It was difficult to conduct further analysis since this error analysis was based on manual categorization. For further investigation, it would be possible to apply Shapley additive explanations (SHAP) [20] or local interpretable model-agnostic explanations (LIME) [21] to visualize the effect of the predictions when the input data is perturbed by deleting or replacing input tokens.

Differences in the distribution of words appearing in the clinical text used for pre-training of UTH-BERT and the pseudo-tweet messages used for evaluation may have affected the errors. Clinical texts differ in that they contain objective information about the patient, while pseudo-tweets message contain subjective information about the tweeter. Another difference is that the former is written in a literary style, while the latter is written in a spoken form. Because of these differences, there may be cases where the representation of the UTH-BERT pre-acquired tokens were not maximally utilized in the pseudo-tweet messages, leading to errors. A limitation of our error analysis is that it was not possible to compare the error trends between the BERT models because we did not perform the error analysis for KU-BERT and TU-BERT. Moreover, given that our developed BERT was evaluated exclusively on the NTCIR-13 MedWeb task, there is currently a limitation in the generalizability of the performance.

3. The comparison between BERT and other classical machine learning methods such as SVM and LR does not make sense in this paper because the superiority of BERT models has been shown in many papers. Furthermore, SVM with KU or TU vocabularies does not make sense because these (subword) vocabularies are not for SVM nor LR.

Response to 3

As you commented, the superiority of BERT models has been shown in previous studies. We removed the sentences that mentioned SVM and LR.

4. There are many typos and some misunderstandings for BERT. Please check the followings carefully.

Response to 4

Thank you for your careful attention. We revised our manuscript based on your comments, and the manuscript was subsequently proofread by native English speakers. We believe it is significantly improved from the previous submission.

5. "A subword that starts with "##" represents a subword that is not the beginning of a word."

Response to 5

Thanks for the important comment. We modified the manuscript as follows:

On Page 4, 

A subword that starts with ## represents a subword that is an appendage to another word.

6. - L122: What does "the sequence" mean? It means a word sequence? The word2vec embeddings, for example, are trained from a word sequence, and so this explanation is inaccurate for mentioning the difference between BERT and existing embeddings.

Response to 6

We modified the manuscript to clearly distinguish between BERT and word2vec as follows:

On Page 6,

The novelty of BERT is that it took the idea of learning word embeddings one step further, by learning each embedding vector considering the co-occurrence of words. 

7. - L123: I think the clause "which learns sentence expressions .. between words" is misunderstanding. The self-attention in BERT can learn token embeddings as well as sentence embeddings.

Response to 7

We modified the manuscript as follows:

On Page 6,

To do this, BERT utilizes the self-attention mechanism, which learns sentence and word embeddings by capturing co-occurrence relationships between those embeddings. 

Response to 8

As you commented, our description was inappropriate. We revised the manuscript based on the original BERT paper (Devlin et al. 2018) as follows:

On Page 6,

This task masks, randomly replaces, or keeps each input token with a certain probability, and estimates the original tokens. Estimating not only the masked tokens but also the replaced or kept tokens help to keep a distributional contextual representation of every input token.

9. - L142: I think the verb "pinch" is not appropriate in this context

10. - L143: I think there is a misunderstanding in the NSP task. For a sentence in a document, a random sentence as a negative example is chosen from other documents. Therefore, it is a usual way that all the sentences in a document in one day are regarded as one document. The explanation from L142 to L147 does not make sense.

Response to 9, 10

Thank you for pointing this out. In response to your suggestion, we deleted the inappropriate expressions and revised the text as follows:

On Page 7,

To develop such a training dataset, for two consecutive sentences in the original dataset, the first sentence is connected to the original second sentence with a probability of 50% as a positive example. The remaining 50% of the time, the first sentence is connected to a randomly sampled sentence as negative example. We treated all sentences appearing in a document recorded in one day for a patient as consecutive sentences.

---

## [Decision Letter · Decision Letter 1]

27 Oct 2021

A clinical specific BERT developed using a huge Japanese clinical text corpus

PONE-D-20-20418R1

Dear Dr. Kawazoe,

We’re pleased to inform you that your manuscript has been judged scientifically suitable for publication and will be formally accepted for publication once it meets all outstanding technical requirements.

Kind regards,

Diego Raphael Amancio

Academic Editor

PLOS ONE

Additional Editor Comments (optional):

Reviewers' comments:

Reviewer's Responses to Questions

**Comments to the Author**

1. If the authors have adequately addressed your comments raised in a previous round of review and you feel that this manuscript is now acceptable for publication, you may indicate that here to bypass the “Comments to the Author” section, enter your conflict of interest statement in the “Confidential to Editor” section, and submit your "Accept" recommendation.

Reviewer #2: All comments have been addressed

2. Is the manuscript technically sound, and do the data support the conclusions?

Reviewer #2: Partly

3. Has the statistical analysis been performed appropriately and rigorously? 

Reviewer #2: Yes

4. Have the authors made all data underlying the findings in their manuscript fully available?

Reviewer #2: No

5. Is the manuscript presented in an intelligible fashion and written in standard English?

Reviewer #2: Yes

6. Review Comments to the Author

Reviewer #2: The manuscript has greatly been improved according to my comments, and

is judged to be acceptable for publication. Although the proposed BERT

model is compatible with other general-domain BERT models, the

presented model and experimental results are valuable for other

researchers especially in the medical domain.

- Minor points:

- L52: a corpus for evaluation -> an evaluation set

- L69: when compared to .. -> compared to ..

- L78: for NLP of Japanese clinical .. -> for Japanese clinical .. (?)

- L82: other -> general

- L114: the sentence -> a sentence

- L250: Experiment settings -> Experimental settings

- L261: I can't understand how the cross-validation was performed. 1

of 4:1 split was used for the development set?

- Table2: Juman -> Juman++

- Table6: The number of categories (13) is relatively large. It is

better to use FP-1, .. FP-8, FN-1, .. , FN-5, and "FP" and "FN"

can be excluded from the interpretations.

7. PLOS authors have the option to publish the peer review history of their article (what does this mean?). If published, this will include your full peer review and any attached files.

Reviewer #2: **Yes: **Tomohide Shibata

---

## [Editor Report · Acceptance letter]

29 Oct 2021

PONE-D-20-20418R1 

A clinical specific BERT developed using a huge Japanese clinical text corpus 

Dear Dr. Kawazoe:

I'm pleased to inform you that your manuscript has been deemed suitable for publication in PLOS ONE. Congratulations! Your manuscript is now with our production department. 

Kind regards, 

on behalf of

Dr. Diego Raphael Amancio 

Academic Editor

PLOS ONE